# CACD-SEG: Contrastive Alignment Consistent Distillation for All Day Semantic Segmentation

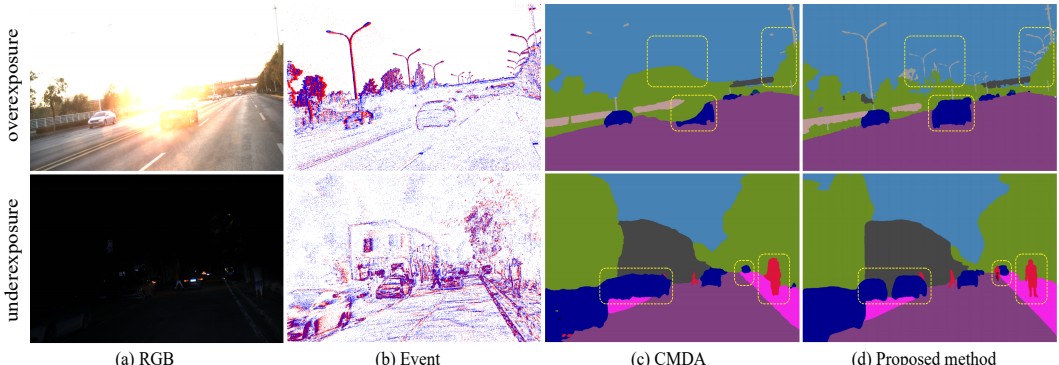

|  |  |  |  |
|---|---|---|---|
| (a) RGB | (b) Event | (c) CMDA | (d) Proposed method |

Figure 1: Segmentation results under extreme lighting conditions. The first and second rows show scenes of overexposure and underexposure respectively. (a) and (b) are paired pixel-aligned RGB images and event frames. (c) and (d) are segmentation results from CMDA Xia et al. (2023) and our proposed method. Clearly, our method performs better by a large margin.

## Abstract

Existing semantic segmentation methods based on frame cameras often encounter issues in complex lighting scenes, such as low-light nighttime or overexposed scenes, and boundary ambiguities caused by motion blur in high-speed scenarios. Event cameras, with their high dynamic range and high temporal resolution, can effectively alleviate these issues and have consequently attracted increasing attention. However, most existing event-based semantic segmentation methods employ straightforward concatenation feature fusion, overlooking the heterogeneity of features between the two modalities. To address these issues, we propose an event-frame alignment-distillation semantic segmentation method. Specifically, we design a heterogeneous feature contrastive alignment module that projects both modalities into a common space to bridge the representation gap. Furthermore, we present a joint boundary-content knowledge distillation module to transfer the clear region and edge information captured by event camera to frame domain, effectively enhancing the robustness of segmentation results. In addition, we construct the first real-world pixel-aligned event-frame semantic segmentation dataset to enable comprehensive training and evaluation, which will be publicly available online. Extensive experiments demonstrate the effectiveness of our method.

## 1 Introduction

Semantic segmentation is a crucial task in computer vision with many significant applications, such as autonomous safe driving Siam et al. (2018) and video surveillance Lin et al. (2018). Although great progress has been made in semantic segmentation under normal lighting conditions Long et al. (2015); Chen et al. (2017); Cordts et al. (2016); Chen et al. (2018); Xie et al. (2021), challenges in extreme lighting scenarios remain unsolved. The low dynamic range and low temporal resolution of frame camera lead to phenomena such as overexposure, underexposure, and motion blur as shown in

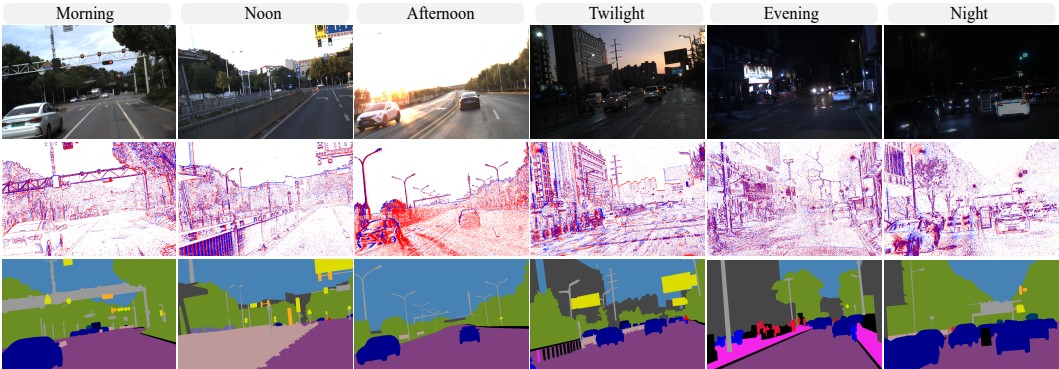

Figure 2: Real-world Pixel-aligned Event-frame All-day semantic segmentation dataset. The first, second and third rows display the RGB images, event frames and annotations respectively, demonstrating the richness of temporal dimensions and lighting conditions in RPEA.

Figure 1(a). This signal-level loss of information cannot be recovered by deep learning techniques. Therefore, solely relying on frame cameras leads to performance drop in segmentation.

To address the limitations of frame camera, previous works Alonso & Murillo (2019) decided to introduce event camera. Event cameras generate the spatiotemporal coordinates of pixels whose luminosity changes exceed a threshold value Finateu et al. (2020). Their unique imaging mechanism provides them with high dynamic range and high temporal resolution Jiang et al. (2023). These characteristics are particularly advantageous in scenarios such as over/underexposure and motion blur, allowing event cameras to provide complementary information for frame cameras. However, event cameras do not capture color information and their spatial data is quite sparse, which limits their performance in segmentation tasks Wang et al. (2021). To this end, we utilize both frame and event modalities to tackle the all-day semantic segmentation challenge.

When addressing all-day event-frame semantic segmentation, two crucial challenges need to be resolved: (i) The absence of real-world, event-frame paired semantic segmentation datasets for driving scenarios. Existing datasets either have synthetic event modality Zhang et al. (2021) or simulated labels generated by pre-trained models Binas et al. (2017); Gehrig et al. (2021b). (ii) Due to the huge domain gap between event and frame representation, how to transfer knowledge from event to frame modality to help improve performance under over/under-exposure and motion blur conditions remains a complex problem. Previous methods Zhang et al. (2021) for dual-modal fusion of event and frame data overlooked the heterogeneity at representation level, resulting in networks that fail to fully exploit information from events.

To tackle the above challenges, we construct a large-scale real-world, event-frame paired semantic segmentation dataset for driving scenarios—RPEA. Specifically, we design a coaxial optical imaging system comprising an event camera and a conventional frame camera, allowing for the simultaneous acquisition of events and images. RPEA contains 4058 image-event pairs which are densely annotated with fine-grained semantic segmentation labels. As illustrated in Figure 2, our dataset exhibits great diversity in lighting conditions and temporal dimensions.

Furthermore, we introduce Contrastive Alignment Consistent Distillation framework (CACD) for all-day semantic segmentation task. Observing the large domain gap between event and frame inputs, we design Heterogeneous Feature Contrastive Alignment module (HFCA) to align the event and frame representation. We leverage the rich semantic knowledge in large model SAM Kirillov et al. (2023) to construct a common semantic space, and then bridge the gap via contrastive learning. Then we design Joint Boundary-Content Distillation module (JBCD) to fully utilize the complementarity of event and frame, smoothly transferring clear boundary knowledge and extreme exposed region knowledge from event to image domain to help segmentation.

In brief, our contributions can be summarized as follows:

- We construct a event-frame semantic segmentation dataset—RPEA. To our best knowledge, RPEA is the first real-world event-frame semantic segmentation dataset with all-day scenarios.

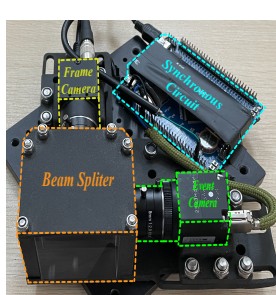 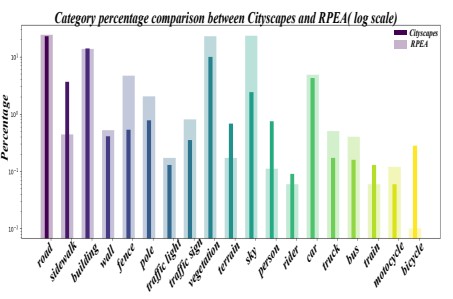 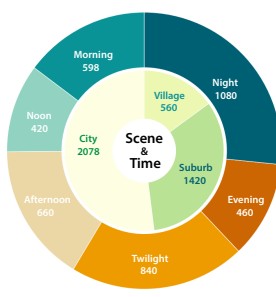

(a) Imaging system          (b) RPEA dataset category distribution          (c) Scene & time distribution

Figure 3: Features of RPEA dataset. (a) Implementation of our coaxial optical imaging system. (b) Distributions of labeled pixels in Cityscapes and RPEA. (c) Scene and time distribution of RPEA.

- We introduce Contrastive Alignment Consistent Distillation framework, contrastively aligning frame and event in a common semantic space to address representational heterogeneity, and then consistently transferring boundary-content knowledge from event to image to improve semantic segmentation performance.

- Extensive experiments over multiple datasets showcase CACD outperforms existing state-of-the-art methods without significantly increasing model parameters.

## 2 RELATED WORK

**Event-based Semantic Segmentation.** Compared to RGB semantic segmentation, event-based semantic segmentation remains underexplored due to the lack of high-quality datasets. Utilizing the paired image-event data in DDD17 dataset Binas et al. (2017), EV-SegNet Alonso & Murillo (2019) used a pretrained image-based network to generate pseudo labels for corresponding events. Since then, labeled events data has been used to train event-based networks in a supervised manner. In addition, ESS Sun et al. (2022) proposed an unsupervised domain adaptation method to transfer knowledge from labeled image datasets to unlabeled event data. CMDA Xia et al. (2023) utilized the gradient of images as a bridge to close the domain gap between event and frame data. Recently, OpenEss Kong et al. (2024) leveraged pretrained large language-vision model CLIP Radford et al. (2021) to transfer knowledge from the image and text domain to event domain to learn a better representation. HybridNN Li et al. (2025) combines SNN and ANN to fuse event and frame to reduce energy consumption. EISNet Xie et al. (2024) filters event streams with attention mask, enhancing salient event features while suppressing noise. BRENet Yao et al. (2025) utilizes optical flow to enhance spatiotemporal alignment of RGB and event. However, existing works did not fully address the semantic consistency and modality heterogeneity between RGB and event, failing to fully exploit the advantages of event.

**Event-Frame Semantic Segmentation Dataset.** Most of the existing event-frame semantic segmentation datasets are synthetic, such as EventScape Gehrig et al. (2021a), DADA-seg Zhang et al. (2021), and DELIVER Zhang et al. (2023b). They use simulators or pretrained networks Zhu et al. (2021) to generate event modality, which differs significantly from real-world events. Another set of datasets like DDD17 Binas et al. (2017) and DSEC Gehrig et al. (2021b) record real-world events, but their semantic labels are generated by pretrained networks, resulting in poor quality. The DSEC Night-Semantic dataset Xia et al. (2023) contains 150 manually annotated labels, which are insufficient for training purpose and only suitable for testing. Our RPEA dataset fills the gap of the large scale real-world event-frame semantic segmentation dataset.

## 3 RPEA DATASET

**Coaxial Optical Imaging System.** Semantic segmentation is a dense prediction task in which each pixel must be assigned a class label. Therefore, achieving pixel-level alignment between the event and frame modalities at the image level becomes crucial. To this end, we ensure pixel-level alignment from both hardware and algorithmic perspectives. First, we construct a coaxial optical imaging system, including an event camera (Prophesee EVK4, 1280*720), a frame camera (FLIR

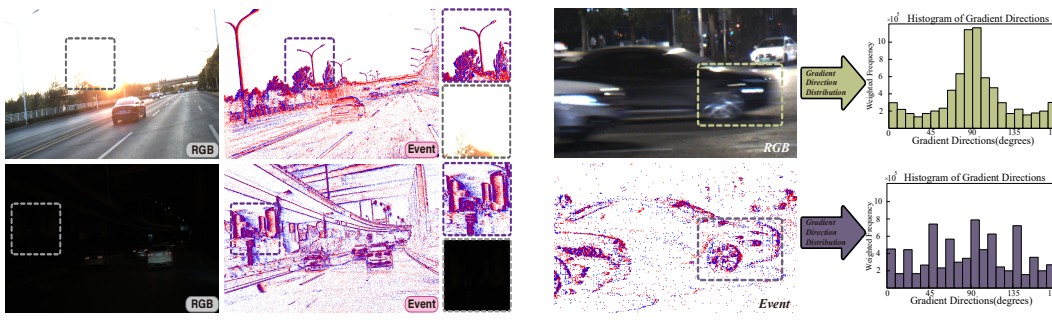

(a) Extreme-exposed scenarios          (b) Motion blur analysis

Figure 4: Advantages of event camera. (a) Event cameras exhibit robustness to lighting variations with high dynamic range. (b) RGB's gradient directions histogram shows strong alignment near 90°, implying severe horizontal motion blur in image. Event's distribution is smoother, less affected by motion blur, exhibiting sharper boundaries with high temporal resolution.

BFS-U3-32S4C, 2048*1536) and a beam splitter (Thorlabs BSW26R), as illustrated in fig. 3(a). The beam splitter divides the incoming light into two equal parts, directing them respectively into the event camera and frame camera. Additionally, we build a programmable synchronous circuit to provide external trigger signals to the cameras, ensuring synchronization of their timestamps. Ultimately, we achieve pixel-level alignment between two cameras through the stereo rectification.

**Annotations.** Annotating RPEA dataset poses greater challenges compared to a traditional image-based segmentation dataset. The dataset comprises two different modalities and contains many complex lighting scenarios such as overexposure, underexposure and motion blur, which greatly increase the difficulty of segmenting objects. To improve the accuracy and reliability of ground truth, we present RGB images, event frames and an overlay of both modalities side by side to the professional annotators, synchronizing their annotation traces to provide useful reference information. This arrangement helps the annotators to label the less visible objects.

**Statistical Analysis.** In fig. 3(b), we compare the distribution of labeled pixels between RPEA and Cityscapes. Since some categories have significantly more pixels than others, we present the distribution in a log scale. Overall, the distribution of most categories in all-day scenarios is similar to daytime scenarios, except for a few specific categories such as sky and bicycle. As demonstrated in fig. 3(c), our dataset exhibits diversity both in terms of temporal dimension and scene dimension.

The RPEA contains 4058 event-image pairs with a resolution of 1034*617. Besides, the RPEA is split into training and validation sets, which consist of 163/40 videos, leading to 3258/800 image pairs. We split them in such way to avoid similar scenes in training and validation sets.

## 4 METHOD

Figure 5 illustrates an overview of our proposed CACD framework. The core of our framework lies in two modules: Heterogeneous Feature Contrastive Alignment and Joint Boundary-Content Distillation. We utilize HFCA to address representation heterogeneity, and leverage JBCD to distill clear boundary knowledge and over/under-exposed region content knowledge from event to image to reinforce the performance in hard regions.

The inputs to the model are paired, pixel-level aligned RGB images and event frames. The event modality consists of a temporally ordered stream of events $\varepsilon_i$, recorded as quadruplets $(x_i, y_i, t_i, p_i)$, which include the coordinates of the pixel $(x_i, y_i)$, a microsecond-level timestamp $t_i$, and a polarity $p_i \in \{+1, -1\}$ indicating an increase or decrease in brightness. We accumulate a period of such event streams to create image-like frames $I_i^{evt} \in \mathbb{R}^{3 \times H \times W}$. Meanwhile, the frame camera outputs colored frames $I_i^{img} \in \mathbb{R}^{3 \times H \times W}$ that are spatially aligned and temporally synchronized with events, where $H \times W$ represents the spatial resolution.

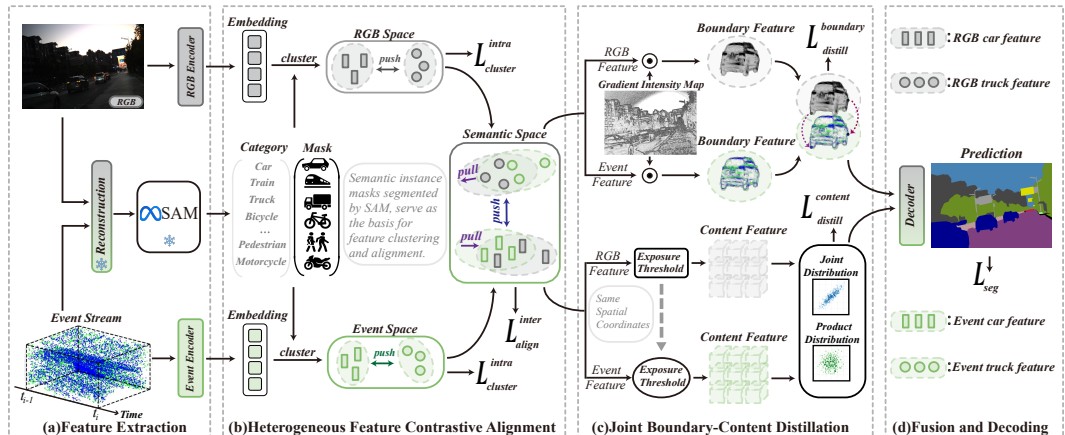

Figure 5: Overview of Contrastive Alignment Consistent Distillation framework. (a) is used to extract features, while (b) aligns the event and RGB features in semantic space. (c) transfers the sharp boundary knowledge and under/over-exposed region content knowledge from event to RGB. (d) fuses features and produces the results.

## 4.1 HETEROGENEOUS FEATURE CONTRASTIVE ALIGNMENT

Our ultimate goal is to leverage the clear knowledge in event and frame to achieve segmentation. However, due to the heterogeneous nature of the image and event inputs, there is a significant domain gap between their representations, which poses great challenges for knowledge transfer. The purpose of the HFCA module is to align their representations in common space to bridge the gap.

Images and events exhibit semantic consistency because pixels at the same coordinate positions in images and events reflect the same semantic concept. Therefore, even though their low-level features differ significantly, they can be aligned at a higher semantic level.

We utilize the rich semantic knowledge in the large model SAM Kirillov et al. (2023) to construct a semantic space, as SAM can generate instance-level masks to indicate regions corresponding to a semantic concept. By aggregating the features of a region indicated by a mask, we can obtain a feature vector that represents an instance. By ensuring that the feature vectors of the same instance in image and event modalities are as similar as possible, and those of different instances are as distinct as possible, we can achieve alignment at the semantic level. Since SAM is trained on clear images, directly applying it to over/underexposed images does not yield good mask results. Therefore, we input both image and event data into an event-frame reconstruction network EvLight Liang et al. (2024) to produce a better-quality image, which is then segmented by SAM to obtain high-quality masks. In practice, we use GroundedSAM Ren et al. (2024) instead of the original SAM.

We utilize contrastive learning to achieve feature alignment. To obtain a better feature space distribution, we approach the task in two steps: intra-domain clustering and inter-domain alignment. The purpose of intra-domain clustering is to maximize the difference between features of different instances within each domain, learning a better intra-domain representation distribution. The goal of inter-domain alignment is to bring features of the same instance in image and event closer, while pushing features of different instances further apart.

First, we use a projection layer to map the original features $f^{img}$ and $f^{event}$ into a common space:

$$F^{evt}_{(i,j)} = P_{evt}(f^{evt}_{(i,j)}), F^{img}_{(i,j)} = P_{img}(f^{img}_{(i,j)}) \tag{1}$$

where $P_{evt}$ and $P_{img}$ are projection layers and $(i,j)$ denotes the spatial coordinate. Then, we use SAM to generate the instance mask $M_k$, corresponding to the $k^{th}$ instance, where $k \in \{1, 2, ..., K\}$, and $K$ is the total number of instances in one image. The value of mask $M_k$ at position $(i,j)$ is 1 if $(i,j)$ belongs to the instance $k$, and 0 otherwise. Then, we use $M_k$ to aggregate the features from the corresponding region, thereby obtaining the feature vector for each instance:

$$F^{evt}_k = \frac{\sum_{i,j} M_k(i,j) \cdot F^{evt}_{(i,j)}}{\sum_{i,j} M_k(i,j)} \tag{2}$$

| Frame | Event | ESS | CMDA | CACD(ours) | GT |

Figure 6: Qualitative results on RPEA dataset. We highlight the details with the yellow boxes.

$$F_k^{img} = \frac{\sum_{i,j} M_k(i,j) \cdot F_{(i,j)}^{img}}{\sum_{i,j} M_k(i,j)} \tag{3}$$

where $F_k^{evt}$ and $F_k^{img}$ denote the event and RGB feature of the $k^{th}$ instance. We achieve intra-domain clustering by penalizing the similarity of features from different instances within each domain to make them more distinguishable:

$$\mathcal{L}_{cluster}^{intra} = \sum_{i=1,i\neq j}^{K} \sum_{j=1}^{K} exp(F_i^{evt} \cdot F_j^{evt}) + exp(F_i^{img} \cdot F_j^{img}) \tag{4}$$

In experiments, we find that this differentiated representation learning significantly enhances the task performance, as segmentation is fundamentally a classification task.

Then we utilize contrastive learning to achieve inter-domain alignment. We pull the image and event features of the same instance closer together, while pushing the image and event features of different instances further apart.

$$\mathcal{L}_{align}^{inter} = -\sum_{i=1}^{K} log \frac{exp(F_i^{evt} \cdot F_i^{img}/\tau)}{\sum_{j=1}^{K} exp(F_i^{evt} \cdot F_j^{img}/\tau)} \tag{5}$$

where $\tau$ denotes the temperature coefficient.

### 4.2 JOINT BOUNDARY-CONTENT DISTILLATION

**Boundary Knowledge Distillation.** Event cameras respond to changes in brightness due to their unique imaging mechanism. This response is particularly strong at the edges of moving objects, resulting in sharp boundary structures. This advantage becomes particularly apparent in scenarios where frame cameras suffer from motion blur as shown in Figure 4(b). Therefore, we decided to transfer the clear boundary knowledge from the event to image, addressing the challenges of motion blur and enhancing the quality of segmentation boundaries.

To transfer the boundary knowledge, we first need to specifically target the boundary features. To achieve this, we compute gradients on high-quality images reconstructed by EvLight, resulting in a gradient intensity map. By multiplying the gradient intensity map with features of images and events, we obtain respective boundary features responding to gradient magnitude. We then enforce a pixel-level consistency loss to transfer the clear boundary knowledge.

$$\mathcal{L}_{distill}^{boundary} = \| F^{img} \odot M_{GIM} - F^{evt} \odot M_{GIM} \|_1 \tag{6}$$

Table 1: Quantitative evaluation on **RPEA**, **CitySca**: Cityscapes, **DSEC-S**: DSEC-Semantic, **DSEC-N**: DSEC Night datasets using mIoU(%). The **best** score is highlighted in **bold**.

| Method | Venue | Modality | Backbone | RPEA | CitySca | DSEC-S | DSEC-N |
|---|---|---|---|---|---|---|---|
| *RGB-based Models* | | | | | | | |
| PSPNet | CVPR'17 | RGB | ResNet-101 | 53.0 | 76.5 | 53.8 | 51.7 |
| OCRNet | ECCV'20 | RGB | ResNet-101 | 54.4 | 78.6 | 53.9 | 53.2 |
| Deeplabv3+ | ECCV'18 | RGB | ResNet-101 | 52.2 | 79.0 | 54.2 | 54.8 |
| SegFormer | NeurIPS'21 | RGB | MiT-B5 | 55.2 | 81.5 | 72.1 | 56.1 |
| SAM | ICCV'23 | RGB | SAM | 55.7 | – | – | – |
| *Event-based Models* | | | | | | | |
| ESS | ECCV'22 | Event | ResNet-18 | 38.2 | 47.3 | 53.3 | 37.6 |
| Ev-SegNet | CVPRW'19 | Event | Xception | 39.1 | 46.9 | 51.7 | 39.2 |
| EvSegformer | TIP'23 | Event | MiT-B3 | 39.7 | 47.5 | 52.1 | 37.3 |
| ESEG | AAAI'25 | Event | MiT-B1 | – | – | 57.5 | – |
| *RGB-Event Models* | | | | | | | |
| BRENet | arXiv'25 | RGB-E | MiT-B2 | 54.5 | 81.2 | **74.9** | 54.5 |
| EISNet | TMM'24 | RGB-E | MiT-B2 | 53.1 | 79.2 | 73.0 | 54.2 |
| CMX | TITS'23 | RGB-E | MiT-B5 | 55.8 | 80.9 | 72.4 | 54.9 |
| CMNeXt | CVPR'23 | RGB-E | MiT-B4 | 54.4 | 80.6 | 72.5 | 55.2 |
| SE-Adapter | ICRA'24 | RGB-E | SAM | – | – | 69.7 | – |
| HybridNN | AAAI'25 | RGB-E | LIF(SNN) | – | – | 66.5 | – |
| ISSAFE | IROS'21 | RGB-E | ResNet-18 | 52.5 | 72.3 | 54.5 | 50.3 |
| CMDA | ICCV'23 | RGB-E | MiT-B5 | 57.2 | 81.9 | 56.3 | 61.2 |
| **CACD** | **Ours** | **RGB-E** | **MiT-B5** | **62.8** | **82.1** | 73.2 | **64.5** |

where $M_{GIM}$ denotes the gradient intensity map. Experiments prove that using $L_1$ loss to constrain boundary features of fine areas is very effective, improving the accuracy of edge differentiation.

**Content Knowledge Distillation.** Event cameras have higher dynamic range compared to traditional frame cameras. This allows them to capture information lost in over/under-exposed areas, thereby providing complementary knowledge as illustrated in Figure 4(a). Therefore, we decide to retrieve the extreme-exposed region knowledge from event and transfer it to image domain.

To transfer knowledge from extreme exposed areas, we need to target the spatial coordinates of regions. Like previous studies Tan et al. (2021), we use the V channel in HSV space to represent exposure intensity, setting two thresholds, $\alpha$ and $\beta$. Areas where the V value is below $\alpha$ are defined as underexposed, and those above $\beta$ as overexposed, identifying the extreme exposed region. The event selects the same spatial coordinates as RGB. By thresholding, we obtain features corresponding to extreme exposure area.

We transform the two content features into probability distributions $P(f^{evt})$ and $P(f^{img})$ within each channel after normalization. The channel count matches the number of categories, ensuring that probability distribution on each channel reflects the spatial distribution of each semantic concept. We calculate the joint and product distributions of $P(f^{evt})$ and $P(f^{img})$, then maximize the KL divergence between the joint distribution and product distribution.

$$\mathcal{L}_{distill}^{content} = -\sum P(f^{evt}, f^{img}) \cdot log \frac{P(f^{evt}, f^{img})}{P(f^{evt})P(f^{img})} \tag{7}$$

Mathematically, by maximizing the KL divergence between the joint distribution and product distribution, we enhance the correlation and information sharing between $P(f^{evt})$ and $P(f^{img})$. This essentially improves the semantic consistency of content features between event and RGB.

### 4.3 LOSS FUNCTIONS AND IMPLEMENTATION DETAILS

Due to the well-learned representations, our fusion process is simple and lightweight. We concatenate the features of event and RGB, then fuse them with multi-head attention mechanism. We utilize the decoder from SegFormer to produce the segmentation results. The overall loss is formalized as:

$$\mathcal{L}_{total} = \mathcal{L}_{CE} + \lambda_1 \mathcal{L}_{cluster}^{intra} + \lambda_2 \mathcal{L}_{align}^{inter} + \lambda_3 \mathcal{L}_{distill}^{boundary} + \lambda_4 \mathcal{L}_{distill}^{content} \tag{8}$$

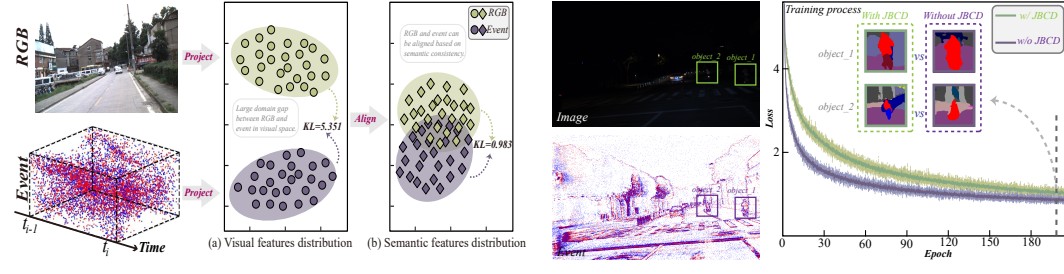

(a) Effect of HFCA module  (b) Effect of JBCD module

Figure 7: Ablation studies. (a) t-SNE visualization of visual and semantic features. HFCA can align event and RGB features in semantic space. (b) JBCD can significantly enhance the performance in underexposed region, making effective use of the corresponding features in the event modality.

where $\lambda_1$, $\lambda_2$, $\lambda_3$, $\lambda_4$ are weighting parameters for balancing the losses. We set $\lambda_1 = 0.0005$, $\lambda_2 = 0.01$, $\lambda_3 = 0.001$, $\lambda_4 = 0.001$ respectively.

Our network is implemented on the Pytorch platform, trained with two Nvidia RTX 3090 GPUs. We adopt MiT-B5 as encoder and Adam optimizer with an initial LR of 0.0001. Images are randomly cropped to the size of $512 \times 512$. We set the batch size to 16 and train for 200 epochs.

## 5 EXPERIMENTS

**Experimental Settings.** We compare various segmentation methods through experiments on RPEA, Cityscapes, DSEC-Semantic and DSEC Night. These include image-based methods (PSPNet Zhao et al. (2017), OCRNet Yuan et al. (2020), Deeplabv3+ Chen et al. (2018), SegFormer Xie et al. (2021)), event-based methods (ESS Sun et al. (2022), Ev-SegNet Alonso & Murillo (2019), EvSeg-Former Jia et al. (2023), ESEG Zhao et al. (2025)), dual-modal methods (CMX Zhang et al. (2023a), CMNeXt Zhang et al. (2023c), HybridNN Li et al. (2025), ISSAFE Zhang et al. (2021), CMDA Xia et al. (2023), EISNet Xie et al. (2024), BRENet Yao et al. (2025), SE-Adapter Yao et al. (2024)).

### 5.1 QUANTITATIVE AND QUALITATIVE EVALUATION

**RPEA dataset.** First, we compare our CACD with other SOTA models. Results in Table 1 reflect the superiority of our method, surpassing CMDA Xia et al. (2023) by 5.6%, proving that we can better extract and utilize the knowledge in event modality. Figure 6 demonstrates significant improvements in segmentation performance brought by our method, especially on the boundaries of objects and in extreme exposed region and high-speed moving vehicles.

**Cityscapes synthetic event dataset.** In ISSAFE Zhang et al. (2021), synthetic events are generated for Cityscapes. In Table 1, compared to the baseline SegFormer, the improvement brought by the event is not significant (+0.6%), which is within expectations. Since event is simulated from video captured by frame cameras, it can't provide truly complementary knowledge at signal level. This also proves the importance of RPEA dataset.

**DSEC-Semantic and DSEC Night.** We train and test on DSEC-Semantic Sun et al. (2022). Besides, we train on RPEA and directly test on DSEC Night dataset Xia et al. (2023) to verify the generalizability of our method. Table 1 demonstrates significant improvements. There is notably a 3.3% increase on DSEC Night, proving the superiority of our method in low-light conditions. Our method does not achieve SOTA on DSEC-Semantic, primarily because its labels are generated by pretrained model and are of limited quality, serving only as a rough reference.

### 5.2 ABLATION STUDY AND DISCUSSION

**Fair Comparison and Model Efficiency.** We conducted a fair comparison with the competing methods, including retraining all methods such as CMDA Xia et al. (2023) on RPEA. We standardized preprocessing, data augmentation, and training epochs for all models. Besides, we only use SAM Kirillov et al. (2023) and EvLight Liang et al. (2024) during training (both frozen). For inference, just the encoder, decoder and fusion are needed, keeping the model lightweight and ensuring

fair comparison. As shown in Table 3, compared with other MiT-based methods, our model achieves a good balance between performance and efficiency without significantly increasing the number of parameters or inference time. The parameters and GFLOPs are counted in $512 \times 512$.

**Importance of event modality.** In table 2, we demonstrate the comparative performance of different input modalities. Adding event provides a 7.6% performance gain over only using image, confirming that event cameras can indeed offer complementary information. However, using only event as input, the performance (39.2%) is far below of using image (55.2%), due to its spatial information being too sparse and lacking color information. This validates the rationality of using a dual-modal approach.

**How does HFCA work?** Table 2 demonstrates the effectiveness of our HFCA module. By contrastively aligning the features of images and events in semantic space, a significant improvement of 3.6% was achieved, indicating that bridging the representation gap can enhance the performance of knowledge transfer and fusion. The large domain gap between event and frame in visual space is bridged in semantic space, as shown in fig. 7a. Besides, HFCA module is very lightweight, as shown in Table 2, adding only 0.3M parameters.

Table 2: Ablation results of different configurations.

| Configuration | | | | Trainable Param (M) | RPEA (mIoU) |
|---|---|---|---|---|---|
| Modality | | Module | | | |
| Image | Event | HFCA | JBCD | | |
| ✓ | ✗ | ✗ | ✗ | 84.6 | 55.2 |
| ✗ | ✓ | ✗ | ✗ | 84.6 | 39.2 |
| ✓ | ✓ | ✗ | ✗ | 174.2 | 57.6 |
| ✓ | ✓ | ✓ | ✗ | 174.5(+0.3) | 61.2(+3.6) |
| ✓ | ✓ | ✗ | ✓ | 174.3(+0.1) | 57.4(-0.2) |
| ✓ | ✓ | ✓ | ✓ | 174.6(+0.4) | 62.8(+5.2) |

**Effectiveness of JBCD.** When knowledge is directly transferred without using HFCA, performance decreases by 0.2%. This is due to the representation heterogeneity between RGB and event, forcibly aligning features at the pixel level can degrade the model's performance. However, when we use the HFCA to project features into semantic space, there is a performance increase of 1.6%, demonstrating that JBCD can effectively distill knowledge and reinforce the performance in tough regions. As shown in fig. 7b, JBCD can significantly improve boundary performance in underexposed area.

Table 3: Model parameters and latency comparison.

| Method | Backbone | mIoU[%] | Trainable Para.(M) | Inference Latency(ms) | GFLOPs |
|---|---|---|---|---|---|
| CMX | MiT-B5 | 55.8 | 181.1 | 32.3 | 143.1 |
| CMNeXt | MiT-B2 | 53.3 | 58.7 | 12.6 | 62.9 |
| CMDA | MiT-B5 | 57.2 | 175.3 | 43.3 | 158.4 |
| Ours | MiT-B5 | 62.8 | 174.6 | 28.3 | 135.5 |

**Is pixel-level alignment really necessary?** In datasets such as DSEC Gehrig et al. (2021b), event camera and frame camera are positioned side-by-side without fully resolving parallax, resulting in misalignment. For dense prediction tasks like semantic segmentation, achieving pixel-level alignment is of paramount importance for two main reasons. First, in annotation process, pixel-level alignment is essential to generate accurate GT. Second, as shown in table 4, we misalign events and RGB images by a certain pixels in random directions, showing pixel-level alignment is crucial for enhancing model performance. This demonstrates the superiority of our coaxial optical imaging system.

Table 4: Impact of misalignment on RPEA using mIoU.

| Time Period | Misalignment(Pixel) | | | | |
|---|---|---|---|---|---|
| | 0 | 1 | 2 | 4 | 8 |
| Morning | 73.5 | 73.2 | 72.9 | 71.5 | 70.9 |
| Night | 55.6 | 51.8 | 51.4 | 50.6 | 50.1 |
| Overall | 62.8 | 60.1 | 59.6 | 58.3 | 58.1 |

## 6    CONCLUSION

In this paper, we construct a real-world pixel-aligned event-frame all-day semantic segmentation dataset—RPEA, featuring many challenging scenarios such as extreme exposure and motion blur. Moreover, we propose the Contrastive Alignment Consistent Distillation framework, addressing the heterogeneity at representation level and then transferring boundary-content joint knowledge based on semantic consistency. The proposed method significantly outperforms the SOTA methods. We believe that our work can contribute to complex scene sensing and parsing.

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
