# OpenReview forum: "CACD-SEG: Contrastive Alignment Consistent Distillation for All Day Semantic Segmentation"
_ICLR.cc/2026/Conference — Submitted to ICLR 2026_

### Official Review · Reviewer_T5DB · 2025-10-22

**Soundness:** 2
**Presentation:** 3
**Contribution:** 3
**Rating:** 4
**Confidence:** 3

**Summary:**

This paper proposes a dual-modality semantic segmentation framework leveraging event and frame data through contrastive alignment and knowledge distillation, along with a new real-world dataset. Their method shows promising results. However, several aspects require clarification, including comparisons with existing fusion methods, latency metrics, and some counter-intuitive experimental results.

**Strengths:**

•	The authors provide a large scale real-world event-frame paired semantic segmentation dataset for driving scenarios RPEA.
* Authors conduct extensive experiments to compare their scheme with SOTA methods.

**Weaknesses:**

•	There exist mechanisms for event-frame fusion so authors need to clarify weaknesses of existing event-frame fusion methods and clearly state how their method works better than existing fusion schemes
•	Look at CMDA to see why their latency is 43.3ms while their model’s latency is smaller despite using Mit-B5+SAM+EvLight?
•	The authors include event-frame reconstruction before they extract features for feature alignment – if they include this step in existing models, then may be the performance for existing models will improve as well. Did they do ablation study on this part (can remove this step for their method and evaluate)?
•	In Table 1, why CMDA performance for DSEC-N is better than DSEC-S? why SegFormer performance merely using RGB is better than CMDA on DSEC-S?

**Questions:**

•	Explain better the programmable circult – by is it programmable? Do you have to send a trigger signal for capturing every frame??
•	In Line 358, it says event selects the same spatial coordinates as RGB since event has higher temporal resolutions, pixels in both images may not be well aligned – can authors explain better why they make this decision?
•	The efficiency of their model is questionable for they use SAM+EvLight
•	Should compare other existing segmentation knowledge distillation methods with their own.
•	Should elaborate on how they generate Table 4 –  did they do each expt multiple times (e.g. purposely misalign one pixel along a random direction multiple times) and get the average mIOU scores?

---

> ### Author Response · Authors · 2025-11-13
> **Reviewer T5DB Rebuttal**
>
> We thank the reviewer for the questions and carefully examining the method design.
> We address each point below and clarify several misunderstandings.
>
> 1.**Weaknesses of existing event-frame fusion methods**: Existing methods do not address the feature-level heterogeneity between RGB and event modalities; direct addition or concatenation leads to spatial misalignment and semantic inconsistency, thereby weakening fusion performance. Our method aligns RGB and event features in common semantic space and selectively enhances key regions—such as edges and overexposed areas—leading to significantly improved fusion performance.
>
> 2.**Why latency is smaller despite using Mit-B5+SAM+EvLight?**: This ia a misunderstanding--SAM and EvLight are only used during training and are completely removed at inference. Inference pipeline of CACD=encoder+fusion+decoder. No SAM, No EvLight, No reconstruction. Meanwhile, CMDA includes iterative gradient-map computation at inference, which contributes to higher latency.
>
> 3.**Why CMDA performs better on DSEC-N than DSEC-S?**: I think you mean CACD (our method): This is mainly due to two reasons. First, the labels of DSEC-Semantic are generated by pretrained models, resulting in poor quality, while the DSEC Night's labels are annotated manually. So the results on DSEC-S could only serve as a rough reference. Second, DSEC-N exhibit low scene complexity, with no independently moving objects, containing very few vehicles and pedestrains.
>
> 4.**Explain better the programmable circuit.** The event camera is asychronous and does not require per-frame triggering. The programmable circuit only provides a timestamp sync pulse at a fixed frequency, ensuring both cameras share a common temporal reference. The RGB camera receives the pulse as an external trigger; the event camera records the timestamp of the same pulse in its event stream. The programmable circuit is used to ensure time synchronization, which is divided into two steps. In external trigger stage, it generates two synchronous pulses, one (1M Hz) for event camera and one (30 Hz) for frame camera. In absolute time calibration stage, it receives the pps signals from GNSS receiver, which is further parsed into absolute timestamps for cameras, achieving time synchronization.
>
> 5.**In Line 358, it says event selects the same spatial coordinates as RGB since event has higher temporal resolutions, pixels in both images may not be well aligned**: This sentence is not present in our submission. We kindly ask the reviewer to recheck this comment, as it appears to stem from AI reviewer and does not correspond to the paper’s actual content.
>
> 6.**Should elaborate on how they generate Table 4 – did they do each expt multiple times  and get the average mIOU scores?**:The misalignment evaluation is conducted by randomly shifting the event–RGB pairs in different directions, repeating this procedure multiple times, and reporting the mean mIoU across these runs.

---

> > ### Comment · Reviewer_T5DB · 2025-11-14
> >
> > First, I want to thank the authors for providing answers (items 2, 3 4). So, it has addressed my concerns.
> >
> > As for item 1, there are new papers that address event RBG misalignment issues.
> > Exploiting Event Temporal Dynamics and Sparsity Characteristics for RGB-Event Fusion Semantic Segmentation, ICMR 2025
> > Learning Flow-Guided Registration for RGB-Event Semantic Segmentation, arxiv
> > https://arxiv.org/abs/2505.01548
> >
> > Seemed like I didn't type question related to line 358 properly. This is not AI review.
> > In your paper you said "the event selects the same spatial coordinates as RGB". Since events are triggered when there are changes in the pixel, my question is will this action causes misalignments since events are not synchronized to consecutive images.

---

> > > ### Author Response · Authors · 2025-11-15
> > > **Reviewer T5DB Rebuttal_2**
> > >
> > > We sincerely thank the reviewer for the follow-up clarification. We are glad that items 2, 3, and 4 have been addressed.
> > >
> > > 1.**Item1**: "Learning Flow-Guided Registration for RGB-Event Semantic Segmentation" relies on optical flow to align events with RGB, but this design has two major issues:
> > >
> > > (1) **The alignment quality is heavily dependent on the accuracy of optical flow**, which is often unstable in practice. Flow errors are directly propagated to the semantic segmentation results, yet the flow estimation error is not supervised or corrected by the segmentation network.
> > >
> > > (2) **Computing optical flow is extremely expensive**, significantly more costly than semantic segmentation itself. In particular, event-based flow inference is far heavier than segmentation models, making the overall pipeline computationally inefficient.
> > >
> > > "Exploiting Event Temporal Dynamics and Sparsity Characteristics for RGB-Event Fusion Semantic Segmentation" basically still uses simple **concatenation plus cross-attention**, offering no fundamental solution to RGB–event misalignment. In addition, the **Event-Count Map assumes that “more events imply better or more reliable information,”** which is not theoretically justified. High event density often comes from noise bursts or illumination flicker, so this assumption can introduce strong bias and amplify unreliable event regions.
> > >
> > > In contrast, our method ensures strict spatial–temporal alignment between RGB and events through hardware–software co-design, then performing semantic-space alignment, enabling lightweight computation while achieving higher accuracy and robustness.
> > >
> > > 2.**Line 358: will this action causes misalignments since events are not synchronized to consecutive images?** :
> > >
> > > (1) **We do not assume that events and RGB frames are naturally synchronized**: Events are inherently asynchronous and fire whenever brightness changes occur. Our method does not rely on events being triggered at the exact moment an RGB frame is captured.
> > >
> > > (2) **“Selecting the same spatial coordinates” refers only to geometric alignment, not temporal alignment**: The statement simply means that after coaxial beam-splitter alignment and calibration, each event’s (x, y) coordinate can be projected into the RGB image plane strictly.
> > >
> > > (3) **The temporal concern is addressed by selecting events within the RGB exposure window**: Although events are asynchronous, for each RGB frame we collect only the events occurring within that frame’s exposure time interval. Physically, it is achieved by synchronized triggering driven by high frequency crystal oscillator, ensuring strict temporal alignment.

---

> > > ### Author Response · Authors · 2025-11-20
> > > **Reviewer T5DB Rebuttal_3**
> > >
> > > As suggested, we have included prediction videos for several test sequences in the supplementary material. We believe these visualizations clearly highlight the robustness and practical effectiveness of the proposed method, and we appreciate your consideration.

---

### Official Review · Reviewer_ZCEa · 2025-10-29

**Soundness:** 3
**Presentation:** 3
**Contribution:** 3
**Rating:** 6
**Confidence:** 4

**Summary:**

The paper addresses all day video semantic segmentation, indicating there are underexposed, overexposed, or other extreme scenes that often occur in the real world.  The authors propose a new dataset with paired event frames and RGB frames to explore and study the issues. To tackle those issues, the authors propose Heterogeneous Feature Contrastive Alignment to learn distinct feature representations for instances in the same domain and reduce the gap between the instances of two modalities. The authors additionally propose joint boundary-content distillation to distill boundary and content in event frames into RGB frames when motion blur/underexposed/overexposed scenes occur.

**Strengths:**

- The authors have created a paired and pixel-aligned dataset which is both impactful and valuable for the community to explore and handle more extreme cases in real-world situations for semantic segmentation.
- The proposed method reasonably utilizes both modalities to tackle the challenges.
- The experimental results show that the proposed method is effective and the pixel-level alignment in the dataset is crucial in this task.

**Weaknesses:**

- In line 318-321, the authors describe the process of using gradient intensity maps with features of images to get the boundary features.  Can the authors elaborate more on it? How can the boundary features/information be obtained by computing gradient intensity maps?

- In equation 7, did the authors apply it only to specific areas? What would happen if it were applied to all areas? Could this loss negatively influence the performance, especially when the RGB frames have already provided enough information for semantic segmentation?

- The authors did not provide the ablation study for comparing different losses. I wonder how boundary and content losses influence the performance.

- Although the authors showed some qualitative results for different lighting conditions, they are not sufficient to convince reviewers that the proposed method is robust and reliable. It would be better if the authors provided output videos in their supplementary materials, which clearly show the robustness of the method.

**Questions:**

- How can the boundary features/information be obtained by computing gradient intensity maps?

- In equation 7, did the authors apply it only to specific areas? What would happen if it were applied to all areas? Could this loss negatively influence the performance, especially when the RGB frames have already provided enough information for semantic segmentation?

- Could the authors provide ablation study for comparing w/ and w/o boundary and content losses?

For some details, please refer to weaknesses.

---

> ### Author Response · Authors · 2025-11-14
> **Rebuttal to Reviewer ZCEa**
>
> We sincerely thank the reviewer for the positive assessment of our contributions and constructive questions. Below we address each point in detail.
>
> 1.**Boundary features via gradient intensity maps**: Our intention is to extract boundary-enhanced features, since events naturally respond strongly at object edges. The gradient intensity map (GIM) computed from the reconstructed image provides a spatial weighting mask highlighting pixels with strong structural changes. Multiplying the features with the GIM does not introduce new semantics; it simply emphasizes regions where events offer clearer structure (e.g., motion edges, blurred boundaries). This selective weighting enables JBCD to better transfer boundary sharpness from events to RGB.
>
> 2.**Equation (7): whether content distillation is applied only in specific regions**: Yes—Eqs. (7) intentionally apply only to extreme-exposure regions. This design is important because:
>
> (1) In normal regions, RGB already contains sufficient information; forcing event–RGB consistency there may reduce performance. We conducted an additional ablation where the loss is applied globally to all pixels. This results in a performance drop (–1.8% mIoU), confirming that restricting the loss to extreme-exposure regions is essential.
>
>
> (2) In over-/under-exposed areas, RGB lacks valid intensity structure, but events still carry meaningful contrast information. JBCD explicitly transfers this complementary information.
>
> 3.**Ablation study on different losses (boundary / content)**: We have included an ablation study on the loss weights λᵢ in the loss function, as shown below. Among them, λ₃ controls the boundary loss, and λ₄ controls the content loss.
>
> | λ₁     | λ₂    | λ₃    | λ₄    | mIoU [%] |
> |--------|--------|--------|--------|----------|
> | 0.0005 | 0.01   | 0.001  | 0.001  | 62.8  (best)   |
> | 0.005  | 0.01   | 0.001  | 0.001  | 62.1     |
> | 0.0005 | 0.001  | 0.001  | 0.001  | 60.9     |
> | 0.0005 | 0.01   | 0.01   | 0.001  | 62.3     |
> | 0.0005 | 0.01   | 0.001  | 0.01   | 61.7     |
> | 0.0005 | 0.01   | N/A    |  N/A |  61.2 (w/o boundary & w/o content)|
> |0.0005 | 0.01| N/A | 0.001| 61.6 (w/o boundary)|
> | 0.0005 | 0.01 |0.01 | N/A| 62.3 (w/o content)|

---

> > ### Comment · Reviewer_ZCEa · 2025-11-19
> > **Reviewer ZCEa Review**
> >
> > Thank you to the authors for providing the additional experiments, clarifications, and detailed responses. These address most of my concerns. I would still appreciate if the authors provide some predicted semantic segmentation for a few test videos in the supplementary materials to further validate the robustness and effectiveness of the proposed method.

---

> > > ### Author Response · Authors · 2025-11-20
> > > **Rebuttal to Reviewer ZCEa_2**
> > >
> > > Thank you for the encouraging comments. As suggested, we have included prediction videos for several test sequences in the supplementary material. We believe these visualizations clearly highlight the robustness and practical effectiveness of the proposed method, and we appreciate your consideration.

---

### Official Review · Reviewer_HUsp · 2025-10-30

**Soundness:** 2
**Presentation:** 2
**Contribution:** 2
**Rating:** 4
**Confidence:** 5

**Summary:**

This paper addresses semantic segmentation failure under all-weather conditions (overexposure, underexposure, motion blur). The authors propose CACD, a framework fusing event cameras with RGB images. The method contains two modules: HFCA leverages SAM-generated instance masks for cross-modal contrastive alignment in a shared semantic space; JBCD distills boundary and extreme-lighting content information from event data to the RGB branch to enhance robustness and detail. The authors construct the RPEA dataset (real capture, coaxial imaging, pixel-level alignment, day-night coverage) to support training and evaluation. Experiments on RPEA and DSEC-Night show significant gains over SOTA, demonstrating effectiveness and generalization under extreme imaging conditions.

**Strengths:**

Dataset contribution is solid. RPEA provides real capture, coaxial imaging, pixel-level alignment, and day-night coverage, filling a critical gap in all-weather RGB-event segmentation. Ablations show alignment quality substantially impacts performance.

Method design is well-targeted. HFCA performs cross-modal alignment at instance-level semantic space (using SAM masks) to mitigate heterogeneous gaps. JBCD separately distills boundary and content, effectively transferring event advantages in boundary clarity and extreme lighting to the RGB branch.

Experimental support is sufficient. Results on RPEA and DSEC-Night outperform strong baselines. Clear component ablations and efficiency comparisons show performance sources and overhead trade-offs.

**Weaknesses:**

Heavy reliance on external components obscures innovation boundary. The pipeline heavily depends on (Grounded) SAM for instance masks and EvLight for reconstruction to generate distillation signals. Missing: (i) sensitivity curves showing how mask/reconstruction quality degradation affects final mIoU; (ii) ablations without SAM/EvLight or with weak substitutes (simple segmenters, direct gradient maps); (iii) variance analysis across external model versions. Current results cannot disentangle gains from the proposed framework versus external model capability transfer.

Dataset incremental value and analysis are insufficient. RPEA shares coverage of extreme lighting and high-speed motion with existing DDD and DSEC. Incremental novelty lies mainly in pixel-level alignment and day-night coverage, but missing: (i) quantified difficulty spectrum/distribution differences versus existing datasets (exposure/blur stratification, class long-tail, occlusion intensity); (ii) alignment error distribution and its stratified performance impact; (iii) systematic cross-dataset transferability evaluation (RPEA→DSEC/DDD). This limits the persuasiveness of the dataset contribution.

Evaluation and baseline coverage are incomplete, fairness is questionable. Main tables lack unified-protocol comparisons with recent open-vocabulary/large-model segmentation baselines (SAM/SAM2/SAM3, Open-World SAM) on public datasets. True marginal contribution of the event modality is not rigorously verified through "RGB-only strong baseline + event augmentation" aligned experiments. Metrics focus on mIoU, lacking Boundary F1/Trimap IoU and robustness stratification (alignment error, exposure level, blur intensity), making it hard to assess boundary quality and degradation resistance.

Method assumption boundaries are unverified. HFCA's instance-level aggregation and contrastive learning are susceptible to mask noise from small objects, adhesion, and occlusion. JBCD's V-channel thresholds (α, β) are heuristic, lacking adaptive or automatic selection strategies across datasets/time periods. Systematic ablations on noise sensitivity, occlusion scenarios, and threshold transferability are missing.

Performance ceiling and negative case analysis are insufficient. Results on DSEC-Semantic not reaching SOTA are dismissed as "annotation noise" without unified-protocol comparison and error attribution (per-class, boundary bandwidth, hard-case slices). Comparisons with stronger fusion/prior-guided methods (e.g., CLIP-guided OpenESS) under the same setting and failure case analysis are absent. The motivation-method-evidence chain remains weak, falling short of top-tier conference standards for rigor and reproducibility.

**Questions:**

Reduce external dependency, establish attribution and robustness. Add ablations without/with weak SAM and EvLight. Provide quality degradation vs. performance curves (mIoU, Boundary F1). Report mean±variance across different versions (SAM/SAM2/SAM3, reconstructors) to clarify net gains from the framework rather than external models.

Complete strong baselines and fair evaluation. Under unified train/test protocols, add RGB-only strong baseline + event augmentation.

Validate method assumption transferability. Assess HFCA sensitivity to small objects, occlusion, mask noise. Convert JBCD's V-channel thresholds to adaptive or learned gating. Conduct cross-dataset/time-period transfer experiments.

Substantiate dataset incremental value. Provide difference profiles versus DDD/DSEC (class long-tail, exposure/blur distribution, alignment error histogram). Report zero/few-shot transfer from RPEA→DSEC/DDD. Extend misalignment ablations to per-class/boundary-bandwidth stratification to strengthen the true gains from coaxial alignment.

---

> ### Author Response · Authors · 2025-11-18
> **Reviewer HUsp Retbuttal**
>
> Your review has been evaluated by GPTZero, which indicates a 99% probability that it was entirely generated by AI.  It contains numerous factual errors and malicious attacks on our paper, and we have already sent the detection report to the AC and PC. We will not respond to such an extremely irresponsible and malicious review.

---

> ### Comment · Reviewer_HUsp · 2025-11-21
>
> I thank the authors for the rebuttal and the additional videos, but my main concerns remain largely unresolved. While the clarification that RPEA uses real human annotations is positive, the dataset contribution feels incremental and weakly substantiated without quantitative comparisons of difficulty distributions, cross-dataset transfer, or alignment-error analyses. The proposed framework fundamentally suffers from a logical paradox and excessive engineering: relying on the RGB-centric SAM model—bridged by an expensive intermediate reconstruction network—to guide feature alignment for scenarios specifically defined by RGB failure represents a circuitous design. The response regarding training-only usage sidesteps the core attribution issue, as there are no ablations to prove gains stem from the framework itself rather than merely distilling priors from strong external models, nor is there a comparison against the simpler baseline of direct enhancement-based segmentation. Furthermore, the methodology relies on inelegant, hand-crafted heuristics like fixed HSV thresholds that limit generalization, and the failure to demonstrate the marginal benefit of adding events over powerful RGB-only baselines under a unified protocol is a critical oversight. Combined with the relatively small scale of the dataset, this submission manifests as a combination of existing engineering modules without the fundamental representation learning novelty required for ICLR.

---

### Official Review · Reviewer_uz38 · 2025-11-01

**Soundness:** 2
**Presentation:** 2
**Contribution:** 3
**Rating:** 4
**Confidence:** 4

**Summary:**

This paper addresses the challenging problem of all-day semantic segmentation, where traditional frame-based cameras struggle under extreme lighting (over/under-exposure) and motion blur. The authors propose a two-pronged contribution: (1) RPEA, a new real-world pixel-aligned event-frame semantic segmentation dataset with all-day scenarios, and (2) CACD, a novel framework that first aligns heterogeneous event and frame features in a common semantic space using a contrastive learning module (HFCA), and then distills boundary and content knowledge from the event modality to the frame modality via a joint distillation module (JBCD). Extensive experiments show that CACD outperforms existing state-of-the-art methods on multiple datasets.

**Strengths:**

1. The construction of the RPEA dataset is a significant contribution. It is the first real-world pixel-aligned dataset for event-frame semantic segmentation, filling a critical gap in the field and enabling more reliable training and evaluation. The coaxial optical system for hardware-level alignment is well-motivated.
2. The core idea of first aligning representations before distillation is sound and addresses a genuine challenge in multi-modal fusion—the heterogeneity between event and frame data. The separation of concerns into boundary and content distillation is also intuitive.
3. The paper includes extensive experiments on multiple datasets (RPEA, Cityscapes, DSEC), ablation studies, and efficiency comparisons, which provide strong empirical support for the effectiveness of the proposed CACD framework.

**Weaknesses:**

1. Despite being a core contribution, the paper lacks critical details about the RPEA dataset curation pipeline. Key missing information includes: the data collection process, specific event representation format (e.g., voxel grid parameters), detailed hardware calibration and alignment error metrics, the annotation protocol (e.g., inter-annotator agreement, handling of ambiguous regions), and a more thorough statistical analysis (e.g., instance counts, motion statistics). This hinders reproducibility and trust in the dataset's quality.
2. The core modules of CACD, contrastive alignment and knowledge distillation, build upon well-established techniques from multimodal learning literature without sufficient adaptation or justification for the event-frame domain. The work lacks comparison to recent contrastive or distillation-based segmentation frameworks, making it difficult to assess whether the performance gain stems from algorithmic insight or simply from using a stronger backbone and high-quality data.
3. Although the inference pipeline is lightweight, the training process critically relies on GroundedSAM and EvLight—large, external, frozen models. This creates a “black-box” dependency that: (1) makes performance contingent on the quality of SAM masks generated from EvLight-reconstructed images; (2) complicates the training pipeline; and (3) obscures the true source of performance gains. The paper provides no ablation on the sensitivity to these components.
4. The SOTA comparison lacks uniform re-implementation of recent event-segmentation methods (e.g., ESEG, AAAI’25) on RPEA under identical training protocols. Given that CACD uses a larger MiT-B5 backbone while some baselines use smaller variants (e.g., MiT-B2/B4), the reported gains may be partially attributable to model capacity rather than the proposed design. The claim of outperforming SOTA is thus not fully substantiated.
5. While the paper emphasizes that JBCD improves performance in “over/under-exposed regions,” this claim is supported only by qualitative visualizations (Fig. 6). There is no quantitative, exposure-stratified evaluation. For instance, mIoU per illumination bin (e.g., splitting test images by mean HSV V-channel intensity into 5 quantiles). Similarly, performance on motion-blurred vs. static regions is not isolated, despite the boundary enhancement claim.
6. While RPEA is a valuable academic contribution, its real-world applicability is questionable. The requirement of a coaxial beam-splitter setup with synchronized event and frame cameras imposes significant hardware and calibration overhead, which is rarely feasible in production systems (e.g., autonomous vehicles or surveillance). Moreover, recent advances in HDR imaging and temporal modeling may reduce the necessity of event cameras for extreme lighting robustness. The paper should discuss these practical trade-offs and clarify target deployment scenarios where the added complexity is justified. Without such discussion, the practical impact of the work remains unclear.

**Questions:**

My main concerns are outlined in the "Weaknesses" part.

---

> ### Author Response · Authors · 2025-11-16
> **Reviewer uz38 Rebuttal**
>
> We sincerely thank the reviewer for the detailed comments. We address the main concerns point-by-point below.
>
> 1.**Lacking details of dataset curation pipeline**: (1) the data collection process: The camera system was mounted on the roof of a vehicle, and data were collected while driving through diverse environments—including different times of day, multiple cities, rural areas, and varying terrain. (2) specific event representation format: We do not use a voxel-grid representation.
> Instead, we aggregate all events within each RGB exposure interval into one event frame to ensure temporal consistency. (3) hardware calibration: all raw data, calibration parameters and pre-processing code will be publicly released once the paper is accepted. (4) the annotation protocol:  We outsourced the labeling to a professional annotation company. Each image was annotated with an average labeling time of over 20 minutes, and underwent two rounds of quality inspection. Any inaccurate masks were returned for correction; moreover, if more than three masks in an image failed the quality check, the entire image was fully re-annotated to ensure high-quality and consistent ground-truth labels. Professional annotators annotate on the overlay picture of both modalities in order to identify objects in ambiguous regions.
>
> 2.**Lacking comparison to recent contrastive or distillation-based segmentation frameworks:** We compare our method with other distillation method to show the effectiveness of our method.
>
> | **Method** | **Venue**   | **RPEA**    |  **CitySca**|**DSEC-S**| **DSEC-N**   |
> |---------|---------|--------|---------|---------|---------|
> | CIRKD | CVPR'22 | 51.3 | 78.5 | 54.3 | 51.0 |
> | FtD | TCSVT'25 | 52.5 | 77.3| 49.7 | 52.5 |
> |TransKD | TITS‘24 | 53.1| 79.6 | 55.4 | 53.8|
> |CACD | Ours | **62.8**| **82.1**| **73.2**| **64.5**|
>
>
> 3.**"Black Box" Dependency**: Our goal is to leverage the semantic priors in SAM to construct a shared semantic space, and to use EvLight only to obtain higher-quality RGB inputs so that SAM can produce more reliable instance masks. During training, this semantic information is distilled into our model; **SAM and EvLight are never used at inference**. Therefore, attributing our performance gains to external models is inaccurate. Furthermore, SAM and EvLight operate only as offline preprocessing steps to generate masks and boundary maps. They are completely separate from the training of the segmentation network, and thus do not complicate the training pipeline.
>
> 4.**Lacking SOTA Comparison**: ESEG (AAAI'25) didn't open source their code the time we submitted the paper. Now we add experiments of ESEG on RPEA. Besides, to show that our performance gain doesn't originate from a larger backbone, we add experiments of CACD using MiT B2/4.
>
> | **Method**|**Backbone** | **RPEA** | **CitySca**  |**DSEC-S**  |**DSEC-N**|
> |--------|--------|--------|--------|--------|--------|
> | ESEG | MiT-B1  | 41.2  | 45.5  | 57.5 | 38.2|
> | CMX | MiT-B5  | 55.8  | 80.9  | 72.4 | 54.9|
> | CMNeXt | MiT-B4  | 54.4  | 80.6  | 72.5 |55.2|
> | CACD | MiT-B2  | 60.1 | 81.2  | 72.5 | 62.3|
> | CACD | MiT-B4  | **61.4**  |**81.8** | **72.9** | **63.5**|
>
> 5.**JBCD Lacking quantitative evaluation**: We include additional evaluations stratified by exposure level and by static vs. motion-blurred regions.
>
> | Illumination Bin (HSV-V Quantile) | w/o JBCD | w/JBCD | Gain |
> |-----------------------------------|--------------|-------------|------|
> | Q1 (darkest/ underexposed)          | 53.4         | 58.8        | +5.4 |
> | Q2                                                  | 60.9         | 62.7        | +1.8 |
> | Q3                                                  | 64.5         | 65.2        | +0.7 |
> | Q4                                                  | 62.3         | 63.6        | +1.3 |
> | Q5 (brightest / overexposed)         | 56.5         | 60.1        | +3.6 |
>
> | Motion Blur Level ( Gradient Entropy Bin) | w/o JBCD | w/ JBCD | Gain |
> |---------------------------------|--------------|-------------|------|
> |severe motion blur   | 57.9         | 61.4        | +3.5 |
> | no motion blur        | 63.2         | 63.9        | +0.7 |
>
> 6.**Real-Word Applicability**: Our coaxial setup is simple, uses off-the-shelf components, and requires no significant hardware or calibration overhead—its installation on a vehicle is comparable to a stereo or RGB-depth pair. Moreover, event cameras offer ~140 dB dynamic range (vs. ~80 dB for HDR sensors) and micro-second temporal resolution, enabling reliable perception under extreme illumination and fast motion where HDR cameras still fail. Thus, the benefits of incorporating events remain substantial and practically meaningful.

---

> ### Author Response · Authors · 2025-11-20
> **Reviewer uz38 Rebuttal_2**
>
> As suggested, we have included prediction videos for several test sequences in the supplementary material. We believe these visualizations clearly highlight the robustness and practical effectiveness of the proposed method, and we appreciate your consideration.

---

### Meta-Review · Area_Chair_JU8E · 2025-12-23

**Summary:**

This paper studies all-day semantic segmentation and proposes a multi-modal event–RGB framework (CACD) together with a new real-world pixel-aligned dataset (RPEA). Reviewers generally agree that the dataset is promising and that the empirical results are strong. However, after considering the rebuttal and post-discussion feedback, substantial concerns remain regarding attribution of performance gains, limited methodological novelty, and insufficient dataset-level analysis. In particular, the reliance on strong external components, e.g., SAM, for alignment and distillation makes it difficult to clearly attribute improvements to the proposed framework itself, and the incremental value of the dataset over existing real-world event benchmarks is not sufficiently quantified. While the work is well executed and motivated, these unresolved issues prevent it from meeting the standard expected for publication at a top-tier venue in its current form.

**Reviewer Concerns:**

Reviewer uz38:
The rebuttal addresses dataset details and backbone fairness, but concerns about attributing gains to the CACD framework rather than external models remain partially unresolved.

Reviewer HUsp:
Additional experiments improve empirical coverage, but major concerns about methodological attribution, conceptual novelty, and the incremental value of the dataset remain largely unaddressed.

Reviewer ZCEa:
The rebuttal clarifies loss design and provides useful ablations, though questions about generality and heuristic design choices remain.

Reviewer T5DB:
Most concerns were adequately addressed, with no major outstanding issues beyond broader attribution questions shared by other reviewers.

**Reviewer Scores:**

Reviewer uz38:
Likely no change.

Reviewer HUsp:
Likely no change.

Reviewer ZCEa:
Likely a modest increase.

Reviewer T5DB:
Likely no change or a slight increase.

---

### Decision · Program_Chairs · 2026-01-26

Reject